# High Mobility Group A1 Chromatin Keys: Unlocking the Genome During MPN Progression

**DOI:** 10.3390/ijms26052125

**Published:** 2025-02-27

**Authors:** Linda M. S. Resar, Li Z. Luo

**Affiliations:** Departments of Medicine (Hematology), Oncology, Pathology and Institute for Cellular Engineering, The Johns Hopkins University School of Medicine, Baltimore, MD 21205, USA; li.luo@jhmi.edu

**Keywords:** High Mobility Group A1 (HMGA1), epigenetic regulator, MPN progression, chromatin keys, transcriptional networks

## Abstract

Patients with chronic, indolent myeloproliferative neoplasms (MPNs) are at risk for transformation to highly lethal leukemia, although targetable mechanisms driving progression remain elusive. We discovered that the *High Mobility Group A1* (*HMGA1*) gene is up-regulated with MPN progression in patients and required for evolution into myelofibrosis (MF) or acute myeloid leukemia (AML) in preclinical models. *HMGA1* encodes the HMGA1 epigenetic regulators that modulate the chromatin state during embryogenesis and tissue regeneration. While *HMGA1* is silenced in most differentiated cells, it becomes aberrantly re-expressed in *JAK2* mutant (*JAK2-V617F*) MPN, with the highest levels after transformation to secondary MF or AML. Here, we review recent work highlighting HMGA1 function in MPN progression. Though underlying mechanisms continue to emerge, increasing evidence suggests that HMGA1 functions as a “chromatin key” required to “unlock” regions of the genome involved in clonal expansion and progression in MPN. Together, these findings illuminate HMGA1 as a driver of MPN progression and a promising therapeutic target.

## 1. Introduction

### Nuclear Architecture, Aging, Clonal Expansion, and Myeloproliferative Neoplasms (MPNs)

Abnormal nuclear shape and chromatin architecture are hallmarks of cancer and the primary feature distinguishing a normal white blood cell from a leukemic cell histologically [1,2,3,4,5]. While cancer cells harbor mutations, mutations alone are insufficient for malignant transformation, and most mutated cells do not give rise to tumors. In keeping with this, the past two decades have witnessed an explosion in studies focused on modifications to chromatin that are independent of DNA sequence—the *epigenome*—launching a new era of epigenomic research. Such work has served to illuminate the central role of the chromatin state in cell fate decisions and cell behavior, both of which are fundamental not only to oncogenic transformation, but also to normal development [3,4,5,6,7,8,9,10,11,12,13]. For example, ground-breaking work in embryonic stem cells (ESCs) and the cellular reprogramming of somatic cells into induced pluripotent stem cells (iPSCs) suggests that accessible, “poised” chromatin facilitates the activation of diverse gene networks, thereby endowing stem cells with long-term self-renewal, plasticity, and other defining “stemness” properties [6,7,8,9,10,11,12,13,14]. Furthermore, nuclei within stem cells and cancer cells tend be larger, with more relaxed, “open” chromatin states compared to differentiated cells. Intriguingly, nuclei also enlarge with aging [15,16]. These distinct settings of large, “open nuclei” could enhance chromatin accessibility to facilitate the expression of genes that foster stem cell properties, such as clonal expansion. In support of this idea, hematopoietic stem cell (HSC) frequency increases with age, and concurrently, HSCs acquire somatic mutations with each cell division [17,18,19,20,21,22,23,24,25,26,27,28]. When a mutation enhances stem cell fitness, or the capacity to self-renew, such as with *JAK2-V617F*, this can result in expansion of the mutant clone, or clonal hematopoiesis, and the development of MPNs [17,18,19,20,21,22,23,24,25,26,27,28,29,30,31,32,33,34,35]. Indeed, aging is the *single most important risk factor* associated with all types of cancer, and aging-related alterations in nuclear structure could provide an optimal chromatin state to initiate MPNs and other forms of clonal hematopoiesis from mutant HSCs. However, not all individuals with *JAK2-V617F* develop MPNs and the time interval required for a mutant HSC to generate a clinical phenotype varies markedly [20,21,22]. When and why MPNs develop in some individuals with *JAK2* mutant HSCs remains a critical unanswered question in the field with obvious clinical import.

## 2. Background

### 2.1. Chromatin Binding Proteins and Nuclear Structure in MPNs—Current Status of the Field

Nuclear proteins accomplish the remarkable feat of compacting eukaryotic DNA by more than 100,000-fold, condensing over 2 m of DNA to less than 20 micrometers [4,5]. Together, DNA and nucleoproteins comprise chromatin, with histones as the most abundant chromatin binding proteins. Histones are positively charged, basic proteins that compact DNA by forming octamer “spools” around which negatively charged DNA fibers wrap [1,2,3,4]. High Mobility Group (HMG) proteins constitute another class of abundant, positively charged chromatin binding proteins with some features in common with histones [36,37,38,39,40,41]. However, HMG proteins have lower molecular weights and, therefore, higher mobility through polyacrylamide gel, thus the name, “high mobility group” [36,37,38,39,40,41,42]. HMG proteins are classified into three families: HMGB, HMGN, and HMGA [36,37,38,39,40,41]. While all are basic, small proteins with an acidic carboxyl terminus, each is defined by unique DNA or nucleosome binding motifs [36,37,38,39,40,41]. All modify chromatin structure, although each has distinct features. Briefly, HMGB proteins are characterized by one HMB-box DNA binding domain or more, whereas HMGN proteins bind to nucleosomes via a nucleosome binding domain, and HMGA proteins bind to AT-rich regions of DNA via AT-hook binding domains [36,37,38]. Here, we focus on HMGA proteins and their role in MPNs.

### 2.2. HMGA Chromatin Binding Proteins: HMGA1a, HMGA1b, HMGA2

HMGA chromatin regulators have emerged as key factors that modulate nuclear architecture and maintain the gene expression required for plasticity and other stem cell properties [36,37,38,39,40,41,42,43,44,45,46,47,48,49,50,51]. HMGA1 chromatin regulators are also among the most abundant, non-histone chromatin binding proteins within the nuclei of cancer cells [36,37,38,39,40,41,42,43,44,45,46,47,48,49,50,51,52,53,54,55,56,57,58]. Encoded by the *HMGA1* gene on chromosome 6p21, HMGA1a and HMGA1b protein isoforms result from alternatively spliced mRNA [36,37]. *HMGA1c*, a rare HMGA1 splice variant, is found only in testes, while HMGA2 (previously HMGI-C), is highly homologous to HMGA1, but encoded by a separate gene, *HMGA2*, on chromosome 12q15 [36]. The HMGA superfamily of proteins are named for three AT-hook motifs that mediate binding to B-form DNA at AT-rich regions within the minor groove [37,38,39,40,41]. Studies using diverse approaches, such as gel shift assays [48,59,60], systematic, exponential enrichment (SELEX) for optimal binding sequences [61], and chromatin immunoprecipitation (ChIP) [45,48,49,52,53,54,55,56,57,58] demonstrate that HMGA1 binds DNA with sequence specifically, although there is only one published study to date of ChIP sequencing (ChIPseq) to delineate HMGA1 binding in cells using an unbiased approach [48]. According to nuclear magnetic resonance (NMR) structural analyses, the AT-hook domain appears unstructured or disordered, but it transitions to an ordered, saddle-like structure after binding to DNA and/or protein partners [38,62]. HMGA bends DNA by penetrating the minor groove via the AT–hooks [62,63,64,65]. HMGA proteins also harbor an amino-terminus that is rich in serine and threonine residues with an acidic carboxyl-terminus, and the latter is thought to mediate interactions with other proteins [36,37,38,39,40]. Similar to HMGN, HMGA1 competes with histone H1 proteins for DNA binding in vitro [36,37,38,39,40,59,60]. After displacing histone H1, HMGA is thought to relax or “open” the minor groove, allowing for the binding of transcription factor complexes and the recruitment of chromatin modifiers to modulate gene expression [36,37,38,39,40,59,60]. The sequence homology in plants suggests that HMGA and histones evolved from the same ancestral protein [66]. Accordingly, both histones and HMGA proteins share a subset of post-translational modifications, although their function and expression patterns have since diverged. As such, HMGA proteins participate in a myriad of cellular processes, including normal embryonic development, cell fate decisions, plasticity, regeneration, proliferation, DNA damage responses, senescence, inflammation, immune signaling, and oncogenic transformation [36,37,38,39,40,41,42,43,44,45,46,47,48,49,50,51,52,53,54,55,56,57,58,59,60,61,62,63,64,65,66,67,68,69,70,71,72,73,74,75,76,77,78,79,80,81,82,83,84,85,86,87,88,89]. Thus, it is not surprising that HMGA proteins are intimately involved in normal HSC function, clonal hematopoiesis, and MPN pathogenesis.

### 2.3. HMGA1 in Normal Development, Transformation, and MPN

HMGA proteins were first linked to cancer when they were identified in the highly metastatic and exceptionally proliferative HeLa cervical cancer cells [67]. Subsequent work established *HMGA1/2* genes among the most highly expressed genes in tumors arising from all three germ layers [36,37,38,39,40,41,42,43,44,45,46,47,48,49,50,51,52,53,54,55,56,57,58,68,69,70,71,72,73,74,75,76,77,78,79,80,81,82,83,84,85,86,87,88,89,90,91,92,93,94,95,96,97,98,99,100]. Gene expression studies (via Northern blot analyses, microarray, RNA sequencing, or PCR) and protein analyses (by mass spectrometry, immunohistochemistry, or immunoblots) demonstrate high levels of *HMGA1*/2 in diverse tumors [68,69,70,71,72,73,74,75,76,77,78,79,80,81,82,83,84,85,86,87,88,89,90,91,92,93,94,95,96,97,98,99,100]. The *HMGA1* gene is a frequent member of gene signatures predicting poor prognosis in diverse tumors [47,48,49,52,68,78,86]. *HMGA1* gene expression and protein levels are also abundant in embryonic stem cells (ESCs) [41,44,46,47], as well as diverse tissue-specific adult stem cells, such as intestinal stem cells and HSCs, with low or undetectable levels in mature, differentiated tissues [12,36,48,49,89,90]. *HMGA1* was also identified in a 13-gene signature of transcription factors enriched in human ESCs [47]; importantly, this signature portends poor outcomes in diverse solid tumors (in breast, bladder, and brain cancer). In pediatric acute lymphoblastic leukemia (ALL), *HMGA1* is overexpressed, with the highest levels at relapse (Table 1) [86]. In MPN, *HMGA1* is overexpressed in at least six independent cohorts [48], including MPN CD34^+^ hematopoietic stem and progenitor cells (HSPCs), compared to healthy, age-matched controls [90,91,92,93], peripheral blood mononuclear cells (PBMCs) [91,92], and even platelet transcripts (Table 1) [94]. In studies that include serial samples taken before and after MPN progression, *HMGA1* levels are highest after transformation into MF or AML [48,92]. Recent work comparing *HMGA1* transcripts in single CD34^+^ cells harboring *JAK2^V617F^* shows that *HMGA1* is overexpressed, but only in *JAK2* mutant cells compared to unmutated CD34^+^ HSPCs from the same individual or to CD34^+^ HSPCs from age-matched healthy controls [48]. While our group and others found increases in *HMGA2* with the progression of indolent MPNs to MF [48], *HMGA2* transcripts are typically magnitudes lower than *HMGA1* in patient samples and *JAK2^V617F^* cell lines using RNA sequencing (RNAseq) technologies to assess transcript numbers. Prior studies also identified *HMGA2* germline variants linked to cancer predisposition [101] and acquired genomic lesions in *HMGA1* [102,103] or *HMGA2* [104] linked to myeloid malignancy. A recent study identified genomic deletions in MF patients at chromosome 12q14.3 with loss of the binding site for the microRNA, *MIRLET2*, which normally represses *HMGA2* expression [104]. As a consequence, affected blood cells in these patients overexpress *HMGA2,* which is likely to foster clonal expansion and transformation based on studies of HMGA2’s oncogenic function in other settings [50,80,82,105]. *HMGA1* is also overexpressed in blasts from adults with a high risk or refractory de novo AML [84,85], and *HMGA2* overexpression portends poor outcomes in AML cohorts [96,97]. Together, these studies highlight HMGA proteins as potential biomarkers and rational therapeutic targets in MPNs, hematologic malignancies, and other tumors (Table 1, Figure 1).

### 2.4. HMGA1 Genetic Variants Are Linked to MPN Risk

Similar to many pluripotency factors and other “epigenetic mediators”, *HMGA1* genes are highly overexpressed in diverse cancers, although they are less frequently mutated, particularly within the coding regions [106,107,108,109,110,111]. Translocations, duplications, and amplifications involving *HMGA1* or *HMGA2* have been reported in solid tumors, myeloid malignancies, such as myelodysplastic syndromes (MDS), and other hematologic malignancies [99,100,101,102,103,104], although such events are relatively infrequent. Rare germline *HMGA2* variants were discovered in children with cancer, though the functional significance of these variants is not yet known [101]. In contrast, translocations involving *HMGA2* are common in benign mesenchymal tumors, such as fibroadenomas, lipomas, and angiomas [99,100]. Genetic alterations affecting *HMGA2* and, to a lesser extent, *HMGA1,* are linked to increased human height in genome-wide association studies (GWAS) [107,108]. Polymorphisms in *HMGA1* are also associated with type 2 diabetes [109]. Results from two large GWAS from the UK biobank identified germline lesions in the noncoding regions of the *HMGA1* locus associated with an increased risk for developing clonal hematopoiesis or MPN [110,111]. However, the incidence of these alterations is rare compared to germline variants that enhance the risk for acquiring an MPN driver mutation, such as those involving the *JAK2* locus [110,111,112], or for developing a significant clinical phenotype in the setting of an acquired driver mutation, such as alterations in the gene encoding the *IL6* receptor [113,114,115,116]. The relative paucity of germline lesions that disrupt the *HMGA1* coding region and function suggest that such mutations are not well-tolerated in humans.

### 2.5. HMGA in Normal Hematopoiesis and Aging

While *HMGA1*/2 (*Hmga1/2*) genes are enriched within HSCs and progenitor cells in gene expression studies on humans [12,48,90,117] and mice [118,119,120,121], the specific stem or progenitor cells with the highest levels vary depending on the study and methodology employed (Table 1). Gene expression studies in humans [122] and mice [118] show that *HMGA1/Hmga1* expression declines with aging in HSPCs, which could contribute to decreasing regenerative function with age. HSCs lacking *Hmga2* have impaired fetal hematopoiesis with slower self-renewal rates, whereas *Hmga2* is dispensable for adult hematopoiesis in murine studies [119,120]. However, in studies with adult mice, complete loss of *Hmga2* disrupts regenerative function during stress hematopoiesis [120]. We found that *Hmga1*-deficient HSCs are less regenerative than wildtype stem cells under stress conditions in competitive bone marrow transplantation (cBMT) [121]. Single-cell RNAseq (scRNAseq) studies demonstrate that HMGA1 is required for maintenance of quiescent HSCs, as well as those HSCs with balanced lymphoid and myeloid potential [121]. Further work is underway to elucidate HMGA1 function in stress and steady-state hematopoiesis.

### 2.6. HMGA1 in Leukemia

Transgenic mouse models demonstrate that *Hmga1a/HMGA1* overexpression causes clonal expansion and leukemic transformation in vivo (Table 1) [69]. In the first *Hmga1* transgenic model reported [69], *Hmga1* expression is driven from the *H-2K* promoter and immunoglobulin *µ* enhancer, leading to the overexpression of murine *Hmga1a* in lymphoid cells. These mice develop T-cell clonal expansion by 28 weeks with frank T-cell acute lymphoblastic leukemia by 5–15 months of age [69,123]. The later onset of lymphoid tumors suggests that the acquisition of new mutations is required for leukemic transformation. *Hmga1* transgenics crossed onto a background deficient in the *Cdkn2a* tumor suppressor locus develop early clonal expansion with lymphoid malignancy at an accelerated rate (20 weeks), with death from disease by 24–26 weeks, compared to *Hmga1a* transgenic mice with intact *Cdkn2a* [123], which is consistent with the requirement of additional mutations for leukemic transformation. In the transgenic mouse model and cell-based studies, HMGA1 induces transcriptional networks involved in proliferation/cell cycle progression, inflammation, cell–cell signaling, hematologic development, and stemness in transgenic lymphoid cells [46,55]. Leukemic blasts from children with B-lineage, acute lymphocytic leukemia (ALL) also overexpress *HMGA1* and similar downstream networks to those identified in preclinical models (Table 1) [46,55,86,124]. Moreover, high *HMGA1* levels correlate with decreased time to relapse in pediatric ALL [86,124]. Furthermore, HMGA1 deficiency in leukemia cell lines disrupts diverse leukemogenic properties, including proliferation, clonogenicity, and leukemic engraftment in immunosuppressed mice [48,124]. *HMGA1* is also highly overexpressed in relapsed and refractory adult leukemia, including both lymphoblastic and myeloid leukemias (Table 1) [48,84,85].
ijms-26-02125-t001_Table 1Table 1Summary of Studies Assessing *HMGA1/Hmga1* in HSC, clonal hematopoiesis, MPN, MDS, and leukemia.
SpeciesAlterationsExpressionCell Type(s)Approach/Rationale/[Reference]Functional StudiesHSCHuman (*HMGA1*)ExpressionEnrichedCD34^+^Microarray [12,117], qPCR [12]scRNAseq [48]Gene silencing, proliferation, clonogenecity [121]Mouse (*Hmga1*)ExpressionEnrichedHSPCsscRNAseq[48,121,125]Genetic deletion in mice, cBMT, clonogenicity,5FU, irradiation [48,125]Clonal Hemato-poeisisHumanVariant/Single nucleotide poly-morphismVariantrs116466979rs115447786rs147627829Human PBMCsDNA sequencing [110,111]GWAS [110,111]MPNHumanExpressionOverexpressionCD34^+^, cell linesqPCR, RNAseq GSE183374 [48]Gene silencing, proliferation, clonogenecity, bone marrow implantation (cell lines)[48]CD34^+^Microarray GSE103237 [93]CD34^+^RNAseq GSE144568 [90]PBMCsRNAseq GSE122198 [92]PlateletsRNAseq Analysis [94]CD34^+^RNAseq GSE103237 [91]MouseExpressionOverexpressionLSK cellsqPCR, scRNAseq GSE197942 [48]Genetic deletion (hetero/homozygous)BMT, clonogenicity, exposure to therapy (ruxolitinib) [48]MDS/AMLHuman ExpressionOverexpressionPeripheral blood blastsqPCR [84,85]Patient responses to therapy[84,85]HumanRearrangementRearrangementBone marrow or PBMCsFluorescence in situ hybridizaion (FISH) [102,103], comparative genomic hybridization [102]Not applicableALLHuman ExpressionOverexpressionPatient bone marrow blasts, cell linesqPCR [55], Microarray [86], RNAseq [124],Gene silencing [55,124], proliferation [55,124], clonogenicity [55,124], bone marrow implantation [124]MouseExpressionOverexpressionBone marrow, spleens, and peripheral bloodqPCR, Microarray [46,55]Transgenic mouse models [46,55]

### 2.7. HMGA1 in MPN Progression

MPNs are perhaps the best examples of clonal hematopoiesis in which increasing *HMGA1* overexpression is the sine qua non for disease progression and leukemic transformation [48,125,126]. Caused by mutations that induce hyperactive *JAK*/*STAT* signaling, MPNs manifest as the overproduction of mature blood cells in chronic disease, which varies depending on the MPN phenotype [29,30,31,32,33,34,35,127,128,129,130,131,132,133,134,135,136,137,138,139,140]. Clinical subtypes include the following: (1) polycythemia vera (PV), characterized by erythrocytosis and thrombocytosis, and frequently associated with leukocytosis, (2) essential thrombocytosis (ET), marked by thrombocytosis, and, (3) myelofibrosis (MF), an advanced MPN phenotype defined by bone marrow scarring and fibrosis associated with decreased survival, splenomegaly, extramedullary hematopoiesis, leukocytosis, and thrombocytosis early on in disease evolution, followed by cytopenias, when bone marrow fibrosis is extensive. *JAK2-V617F* is the most common mutation causing MPNs, occurring in >95% of PV and ~50% of ET cases, although *JAK2* variant allele fractions (VAFs) are typically <50% of ET cases [33]. Both ET and PV are associated with chronic indolent phenotypes and a propensity for bleeding and thrombotic complications. *JAK2-V617F* also occurs in ~50% of MF cases, which can arise from PV or ET (secondary MF) or occur de novo (primary MF). In addition to *JAK2-V617F*, mutations in *CALR* and *MPL* also cause ET and MF [127,128,129,130,131,132,133,134,135,136]. While most commonly diagnosed in middle-aged and older adults, there is a growing recognition of MPNs in younger patients with the advent of more widespread genetic testing [139,140,141,142,143,144,145]. In fact, clonal evolution studies estimate the acquisition of *JAK2* mutations decades before clinical diagnosis, and even in utero, in some cases [144]. Furthermore, studies from newborn screening tests revealed the presence of *CALR* mutations in twins at birth who later developed MF in their fourth decade of life [145]. Interestingly, genetic stability in chronic MPNs (ET, PV) appears to be similar to more indolent forms of clonal hematopoiesis (*DNMT3A*, *TET2*), rather than more aggressive myeloid neoplasms (like myelodysplastic syndromes), since the rate of acquisition of new mutations is low and typically related to patient age rather than disease duration [33,137,138]. Thus, in contrast to other myeloid neoplasms, MPN disease duration is measured in decades rather than years. Paradoxically, however, spontaneous transformation into acute myelogenous leukemia (AML) is not only the most severe MPN complication but, in contrast to de novo AML, refractory to the conventional chemotherapy used for AML [33,35,135,136,137,138].

### 2.8. HMGA1 Is Required for JAK2-V617F Leukemogenesis and Resistance to JAK2 Inhibitors in Preclinical Models

Increasing *HMGA1* expression concurrent with progression in MPNs and refractory leukemia suggested a causal role for HMGA1 in MPN progression [48,126]. In support of this, silencing *HMGA1* via CRISPR/Cas9 or short hairpin RNA approaches demonstrates that HMGA1 is required for salient in vitro leukemia phenotypes, including proliferation and clonogenicity, in *JAK2^V617F^* AML cell lines (Table 1) [48]. Moreover, implantation of *JAK2^V617F^* AML cell lines with HMGA1 deficiency in immunosuppressed mice results in decreased engraftment of leukemic blasts in the bone marrow and spleen, together with a lower frequency of circulating leukemia blasts [48]. Intriguingly, *HMGA1* levels increase in leukemic cells that engraft in the bone marrow from the pool of cells with *HMGA1* silencing, suggesting that a specific level of HMGA1 is required for leukemic engraftment and expansion. Furthermore, HMGA1 deficiency in *JAK2^V617F^* AML cell lines enhances responses to the JAK inhibitor, ruxolitinib, disrupting leukemic engraftment and expansion while prolonging survival in recipient immunosuppressed mice [48]. This result is particularly noteworthy, since AML that arises in the setting of MPNs is refractory to all therapy, including ruxolitinib and conventional, cytotoxic chemotherapy (Figure 1).

### 2.9. HMGA1 Induces Transcriptional Networks Involved in Proliferation and Cell Fate in MPN Leukemia

Given these striking results demonstrating a key role for HMGA1 in MPN leukemia, multiomics studies were performed in *JAK2^V617F^* AML cell lines to identify underlying mechanisms, including RNAseq, chromatin immunoprecipitation sequencing (ChIPseq), and chromatin accessibility studies (assay for transposase chromatin sequencing, or ATACseq) [48]. The transcriptional networks governed by HMGA1 in this setting include the activation of gene networks involved in cell cycle progression (E2F, mitotic checkpoint, G2M, and MYC target genes), in addition to cell fate (GATA2 networks). HMGA1 chromatin occupancy was also detected in gene loci regulating cell cycle progression networks and GATA2 networks. Notably, significant HMGA1 occupancy was not detected at target gene loci downstream of cMYC, suggesting that HMGA1 regulates these networks indirectly, possibly through direct activation of cMYC. Indeed, prior studies showed that HMGA1 binds directly to the *MYC* promoter and induces *cMYC* in embryonic stem cells and iPSCs [45] and, conversely, that HMGA1 is induced by cMYC in other settings (Burkitt lymphoma cells, transformed fibroblasts) [105,146]. Chromatin accessibility associated with high HMGA1 in *JAK2^V617F^* AML cell lines includes gene loci involved in proliferation and signal transduction, pathways that are regulated by HMGA1 in diverse tumor settings [48,49]. Intersecting the HMGA1 transcriptome (RNAseq) with HMGA1-mediated chromatin accessibility (ATACseq) unveiled enrichment for GATA2 gene networks, KRAS signaling, and activating histone marks, including histone 3 lysine 4 monomethylation (H3K4me1) or trimethylation (H3K4me3). Although further studies are needed to identify rational approaches to target HMGA1, these results suggest that hypomethylating agents could interfere with HMGA1 function (Figure 1) [48].

Because high levels of *GATA2* are linked to poor outcomes in myeloid malignancies [147,148,149,150,151] and *GATA2* was among the genes most differentially regulated by HMGA1 [48], functional studies were performed to determine whether HMGA1 depends upon GATA2 and downstream networks for leukemic transformation in MPN models. Similar to HMGA1 deficiency, silencing *GATA2* in *JAK2^V617F^* AML cell lines decreased leukemogenic properties in vitro and in vivo, recapitulating most, but not all of the phenotypes observed with HMGA1 deficiency [48]. By contrast, restoring *GATA2* in *JAK2^V617F^* AML cells with *HMGA1* silencing partially rescues leukemia phenotypes, increasing clonogenicity and leukemic engraftment, without restoring proliferation rates in HMGA1-deficient cells. Mechanistically, HMGA1 binds directly to AT-rich sequences near the *GATA2* developmental enhancer (+9.5), enhancing chromatin accessibility and recruiting active histone marks (H3K4me1/3) to induce *GATA2* [48,147]. Most importantly, HMGA1 transcriptional networks, including *GATA2* and proliferation pathways (G2M, mitotic spindle, E2F) are activated in PBMCs or CD34^+^ cells from patients with MF. Furthermore, in matched samples from MF patients whose condition transformed into AML, *HMGA1*, *GATA2*, and other HMGA1 networks are activated further after leukemic transformation, underscoring the significance of HMGA1 as a master regulator in human MPN. Intriguingly, GATA1 proteins are low in human bone marrow from patients with MPN, and Gata1 deficiency in mice causes MF [150,151,152]. Since GATA1 and GATA2 proteins recognize the same DNA binding sites, HMGA1 may foster a “GATA1 low” phenotype by enforcing *GATA2* expression, thereby precluding GATA1 transcriptional function. Another group discovered increased expression of the nuclear factor erythroid 2 (NFE2) transcription factor in MPN patients. In MPN experimental models, NFE2 is also associated with overexpression of *GATA2* [153], further underscoring the role of GATA2 in MPN pathogenesis. Prior work also linked NFE2 mutations to leukemogenesis in MPNs [154,155]. Because leukemic clones could arise in HSCs lacking NFE2 mutations, it was postulated that paracrine effects also promote leukemogenesis in this setting. Indeed, paracrine inflammatory signals are associated with MPN progression and associated sequelae in experimental models and patients [156]. It is possible that HMGA1 cooperates with these mutant NFE2 pathways to induce *GATA2* and leukemic transformation in MPNs. However, since GATA2 function is required for normal hematopoiesis, its potential as a therapeutic target is limited [157].

### 2.10. HMGA1 Is Required for Overproduction of Myeloid Lineages in JAK2^V617F^ Chronic MPN

Given prior studies linking aberrant *HMGA1* expression to refractory leukemia [55,69,84,85,86,123,124,126], its functional role in MPN leukemic transformation was not unexpected. Surprisingly, however, HMGA1 is also required in more chronic, indolent MPN phenotypes in *JAK2-V617F* transgenic mice that closely model human PV. For example, these *JAK2-V617F* transgenic mice develop erythrocytosis and thrombocytosis as adults (10–12 weeks of age) with a gradual onset of splenomegaly and extramedullary hematopoiesis [48,158]. Myelofibrosis and osteosclerosis also occur, but only after 30–40 weeks, although leukemia does not develop in this model [158]. In *JAK2-V617F* transgenic mice, the loss of just a single *Hmga1* allele was sufficient to dampen erythrocytosis and thrombocytosis while preventing splenomegaly, osteosclerosis, and myelofibrosis [48]. Furthermore, *Hmga1* deficiency within the hematopoietic compartment is sufficient to mitigate MPN progression. Similar to *JAK2-V617F* mice with global, heterozygous *Hmga1* deficiency, those with the loss of one *Hmga1* allele restricted to HSCs have decreased erythrocytosis, thrombocytosis, splenomegaly, and fibrosis [48]. Further analyses via scRNAseq demonstrated a marked reduction in the frequency of long-term, *Procr* high HSC, megakaryocyte-biased HSCs, and megakaryocyte–erythroid-biased HSPC, with an expansion in lymphoid-biased HSCs [48,125]. While further studies are needed to dissect the molecular mechanisms underlying HMGA1 in both chronic MPNs and MPN AML, these striking results reveal a critical role for the HMGA1 epigenetic regulator in MPN pathogenesis (Figure 1).

## 3. Discussion

### 3.1. Looking to the Future: Can HMGA1 Be Modulated in Therapy?

Recent work illuminates a novel role for HMGA1 as an epigenetic “key” that “unlocks” regions of the genome required for MPN progression in *JAK2* mutant HSCs [48,125,126]. In MPN leukemia models, HMGA1 is required for engraftment in the spleen and bone marrow and for clonal expansion [48]. Mechanistically, HMGA1 activates transcriptional networks involved in cell cycle progression and cell fate, including the GATA2 master regulator in *JAK2* mutant AML cells [48]. Accordingly, HMGA1 could foster a “GATA1 low” phenotype by enforcing *GATA2* expression to preclude GATA1 function [151,152,153]. Similar to HMGA1, mutant NFE2 activates *GATA2* expression, along with that of other genes (*SCL*/*TAL1*) that could drive leukemogenesis [48,153]. It is possible that HMGA1 cooperates with NFE2 pathways to induce leukemic transformation in MPNs. Thus, further studies examining transcriptomes in larger cohorts of MPN patients will elucidate additional HMGA1-dependent and independent pathways that become “unlocked” during MPN progression. Studies of single nuclear transcriptomes and chromatin accessibility (single-nucleus multiomics) are also needed to define HMGA1 function within varied cell contexts. More recently, we identified pathways that are repressed by HMGA1 in MPN, and dissecting these “locked-down” transcriptional networks could unveil novel therapeutic opportunities [159]. Intriguingly, in this recent work, HMGA1 down-regulated gene expression by disrupting three-dimensional (3D) genome architecture, highlighting the need for further studies of the effects of HMGA1 on chromatin architecture. Technologies such as chromatin conformation capture coupled with sequencing (HiC) to elucidate the effects of HMGA1 on 3D nuclear structure promise to provide further insight relevant to HMGA1 in MPN progression and other contexts. Indeed, recent work using HiC technologies revealed important 3D alterations in embryonic development, T-ALL, and lineage-ambiguous leukemia [13,160,161].

Surprisingly, the loss of just a single *Hmga1* allele is sufficient to dampen the overproduction of red cells and platelets while preventing splenomegaly and progression to MF [48]. This was unexpected, since HMGA1 overexpression has been primarily linked to more advanced, stem-like, and refractory hematologic malignancies and solid tumors. Further analyses of HMGA1 at early stages in the development of *JAK2-V617F* MPNs are likely to elucidate pathways that could be modulated in therapy, ideally to prevent progression to MF or AML. Intriguingly, further analyses of single-cell transcriptomes from *JAK2-V617F* mice with PV show that Hmga1 induces genes activated by interferon alpha signaling, including receptors that activate interferon alpha pathways [125]. If interferon alpha signaling is activated in *JAK2-V617F* mutant HSCs in patients, this could provide a mechanistic basis for a preferential response of *JAK2* mutant HSCs to interferon therapy compared to their *JAK2* wildtype counterparts, although further studies will be needed to confirm these findings.

### 3.2. How Can HMGA1 Function Be Modulated Pharmacologically?

Since HMGA1 is an architectural transcription factor, targeting HMGA1 in the clinic remains a challenge. However, bromodomain inhibitors (BETis) have efficacy in targeting stretch or “super-enhancer” chromatin complexes that activate the cMYC oncoprotein and downstream pathways, suggesting that a similar approach could be used to perturb *HMGA1* expression [162,163,164]. Alternatively, disrupting HMGA1 chromatin complexes with small molecules could be exploited in the clinic, although none exist to date. Recent advances in the protein degraders, proteolysis targeting **c**himeras (PROTACs), suggest that this approach could be harnessed to investigate the cellular effects of rapid HMGA1 degradation and may lead to therapeutic approaches for patients [165,166,167]. Alternatively, further elucidation of the factors activating HMGA1 could reveal therapeutic opportunities. HMGA1 is induced by growth factors and diverse oncogenic mutations, including *JAK2*, suggesting that upstream pathways could be modulated to decrease *HMGA1* expression and function. In addition, the dissection of downstream pathways activated by HMGA1 should unveil actionable mechanisms. For example, in models of pancreatic cancer, we discovered that HMGA1 activates pathways downstream of the fibroblast growth factor, FGF19, and inhibitors of FGF19 and its receptor (FGFR4) are already available in the clinic [52]. Intriguingly, HMGA1 fosters both tumor progression and fibrosis in models of pancreatic carcinogenesis, much like its activity in MPN progression.

## 4. Conclusions

In closing, emerging evidence focused on chromatin regulators, such as HMGA1, illuminates the epigenome as our next frontier in cancer biology and therapeutics. However, a great deal of work is needed to better harness the epigenome to modulate cellular behavior and cell fate decisions. Such work should not only lead to advances in cancer therapy, but also provide clues as to why some cells harboring mutations, such as *JAK2*, fail to drive aberrant clonal hematopoiesis, whereas others lead to MPNs, progressive disease, and lethal leukemia. More broadly, most mutated cells do not develop into cancer cells, and understanding the mechanisms that lead a cell to transform or preclude cancer development requires our urgent attention, particularly since aging populations are increasing globally, along with the number of individuals with mutant cells, clonal expansion, and cancer.

## Figures and Tables

**Figure 1 ijms-26-02125-f001:**
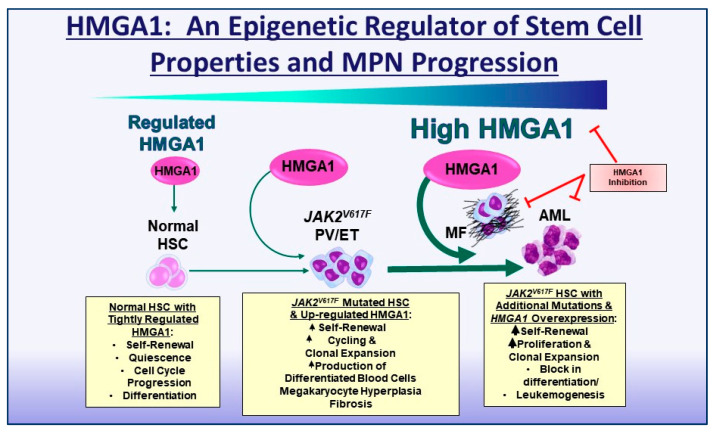
HMGA1 in MPN progression. *HMGA1* is abundant in normal HSCs, where its expression is tightly regulated to maintain appropriate self-renewal, quiescence, and hematopoiesis. When HSCs acquire a *JAK2-V617F* mutation, *HMGA1* becomes up-regulated, whereupon it alters accessible chromatin and induces gene networks involved in self-renewal, clonal expansion, and cell cycle progression, resulting in overproduction of mature blood cells. With aging and acquisition of additional mutations, *HMGA1* is up-regulated further, driving aberrant proliferation and a block in differentiation with leukemic transformation. Developing approaches to perturb HMGA1 function could reprogram the epigenome and prevent clonal expansion and MPN progression.

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
