# Peer review of "High Mobility Group A1 Chromatin Keys: Unlocking the Genome During MPN Progression"

_ijms, 2025, doi:10.3390/ijms26052125_

Round 1
Reviewer 1 Report
Comments and Suggestions for Authors
In the actual manuscript the author provides a comprehensive review on the involvement of High Mobility Group A1 (HMGA1) gene in driving the progression of JAK2 V617F-positive myeloproliferative neoplasms (MPNs). The paper is well written, being covered many aspects concerning leukemogenesis in MPNs, and also possible targeting of HMGA1. An additional scheme to synthesize the data about the pathways regulated by HMGA1 is recommended.
Author Response
Reviewer 1: In the actual manuscript the author provides a comprehensive review on the involvement of High Mobility Group A1 (HMGA1) gene in driving the progression of JAK2 V617F-positive myeloproliferative neoplasms (MPNs). The paper is well written, being covered many aspects concerning leukemogenesis in MPNs, and also possible targeting of HMGA1. An additional scheme to synthesize the data about the pathways regulated by HMGA1 is recommended.
Thank you for the helpful suggestions; we updated the manuscript with a more detailed summary figure and a Table as recommended.
Please disregard the attached file - it is not a final draft of our manuscript and was uploaded before we had access to the other reviews.

Reviewer 2 Report
Comments and Suggestions for Authors
In this current review " High Mobility Group A1 Chromatin Keys: Unlocking the Genome during MPN Progression” the author gives a detailed description of how HMGA1 plays a crucial role in the progression of various cancers and how this information can be used in cancer biology and therapeutics. The review mentions about the important aspects of HMGA1, but these points can be added, and the review can be published.
1) How dynamics of HMGA1 chromatin changes over time has this been reported ever and if not, can it be detected by using some technique.
2) Most of the studies done on HMGA1 are done on the small cohort can this be one of the limitations of this study, and why this study has not been done on larger cohort.
3) In conclusion, authors should also mention the involvement of those techniques that can lead to the use of HMGA1 at the broader aspect.
4) In conclusion, authors should mention also mention a line describing the difference of HMGA1 in aging cells, which would help in better differentiation among them.
Author Response
Reviewer 2: In this current review " High Mobility Group A1 Chromatin Keys: Unlocking the Genome during MPN Progression” the author gives a detailed description of how HMGA1 plays a crucial role in the progression of various cancers and how this information can be used in cancer biology and therapeutics. The review mentions about the important aspects of HMGA1, but these points can be added, and the review can be published.
1) How dynamics of HMGA1 chromatin changes over time has this been reported ever and if not, can it be detected by using some technique. This is an interesting question, although there are no studies to date to address this and the discussion was modified to reflect this (highlighted in yellow).
2) Most of the studies done on HMGA1 are done on the small cohort can this be one of the limitations of this study, and why this study has not been done on larger cohort. Our publication in Blood on HMGA1 in MPN includes JH primary patient samples (n=18 MPN patients with CD34+ cells and gene expression analyzed by qPCR), in addition to gene expression data from 4 additional cohorts, including: 1) a cohort of early stage disease (n=50 MPN patients with bone marrow-derived CD34+ cells and gene expression analyzed by microarray), 2) a cohort of MF patients only (n=8, with single cell CD34+ cells and gene expression analyzed by single cell RNAseq (scRNAseq) comparing mutated versus unmutated cells), 3) another MF cohort (n=15 with CD34+ cells and gene expression analyzed by scRNAseq), and, 4) a cohort of matched MF who transformed to AML (n=11) with RNAseq from peripheral blood mononuclear cells. Thus, while these cohorts are relatively small, these data comprise most of the available gene expression data. Our leukemia cohorts (B-ALL) cited in this review are larger since gene expression in pediatric ALL has been more extensively studied. We added text (highlighted in yellow) to indicate the importance of studying additional patient samples at the single cell level, which are beginning to emerge in the literature.
3) In conclusion, authors should also mention the involvement of those techniques that can lead to the use of HMGA1 at the broader aspect. Thank you for this important point. We added text (highlighted in yellow) designating techniques (HiC, single nuclei multiomics) that will likely shed further light on HMGA1 function.
4) In conclusion, authors should mention a line describing the difference of HMGA1 in aging cells, which would help in better differentiation among them. Thank you for this important point. There is limited work on HMGA1 in aging, although we have mouse models and extensive unpublished results which will be submitted soon. We added text to address this and an additional reference examining gene expression in human CD34+ cells with aging (Prall et al.)
Reviewer 3 Report
Comments and Suggestions for Authors
This is a very interesting review on the role of high-mobility group A1 (HMGA1) DNA binding proteins in normal development and cancer. It is written by a renowned expert in HMGA function in leukemia and it contains relevant and compelling information on the topic. The review article is well-written and exhaustive. A couple of minor adjustments are recommended to improve the overall quality of the manuscript.
- Section 2.3 lists the various roles of HMGA1 in development, transformation and malignancy. Given the density of information, I would advise to insert a table to summarize the findings.
- Introduction, lines 27-29. While listing the role of chromatin architecture in leukemia (mainly B-ALL), the authors may reference also studies on the T lineage (PMID: 32203470 and PMID: 34103329)
- Introduction, line 35. While describing chromatin role in cell fate decision, a recent reference may be added (PMID: 38053013)
Author Response
Reviewer 3: This is a very interesting review on the role of high-mobility group A1 (HMGA1) DNA binding proteins in normal development and cancer. It is written by a renowned expert in HMGA function in leukemia and it contains relevant and compelling information on the topic. The review article is well-written and exhaustive. A couple of minor adjustments are recommended to improve the overall quality of the manuscript.
- Section 2.3 lists the various roles of HMGA1 in development, transformation and malignancy. Given the density of information, I would advise to insert a table to summarize the findings. Thank you – we added a table focused on HMGA1 in normal hematopoiesis and hematologic malignancies.
- Introduction, lines 27-29. While listing the role of chromatin architecture in leukemia (mainly B-ALL), the authors may reference also studies on the T lineage (PMID: 32203470 and PMID: 34103329) Thank you – we added these interesting and relevant references.
- Introduction, line 35. While describing chromatin role in cell fate decision, a recent reference may be added (PMID: 38053013) Thank you for pointing out this interesting reference – we added text referring to it and the reference.